# DexScale: Automating Data Scaling for Sim2Real Generalizable Robot Control

Guiliang Liu [* 1]   Yueci Deng [* 2]   Runyi Zhao [1]   Huayi Zhou [1]   Jian Chen [2]   Jietao Chen [2]   Ruiyan Xu [1]
Yunxin Tai [2]   Kui Jia [1 2]

## Abstract

A critical prerequisite for achieving generalizable robot control is the availability of a large-scale robot training dataset. Due to the expense of collecting realistic robotic data, recent studies explored simulating and recording robot skills in virtual environments. While simulated data can be generated at higher speeds, lower costs, and larger scales, the applicability of such simulated data remains questionable due to the gap between simulated and realistic environments. To advance the Sim2Real generalization, in this study, we present DexScale, a data engine designed to perform automatic skills simulation and scaling for learning deployable robot manipulation policies. Specifically, DexScale ensures the usability of simulated skills by integrating diverse forms of realistic data into the simulated environment, preserving semantic alignment with the target tasks. For each simulated skill in the environment, DexScale facilitates effective Sim2Real data scaling by automating the process of domain randomization and adaptation. Tuned by the scaled dataset, the control policy achieves zero-shot Sim2Real generalization across diverse tasks, multiple robot embodiments, and widely studied policy model architectures, highlighting its importance in advancing Sim2Real embodied intelligence. The project webpage at: https://edem-ai.github.io/dexscale.github.io/.

## 1. Introduction

A key milestone in advancing modern AI systems is the development of embodied intelligence, which aims to seamlessly integrate an agent's physical body, sensory perception, and environment into its learning, reasoning, and decision-

---

[*]Equal contribution  [1]School of Data Science, The Chinese University of Hong Kong, Shenzhen [2]DexForce, Shenzhen. Correspondence to: Kui Jia <kuijia@cuhk.edu.cn>.

*Proceedings of the 42nd International Conference on Machine Learning*, Vancouver, Canada. PMLR 267, 2025. Copyright 2025 by the author(s).

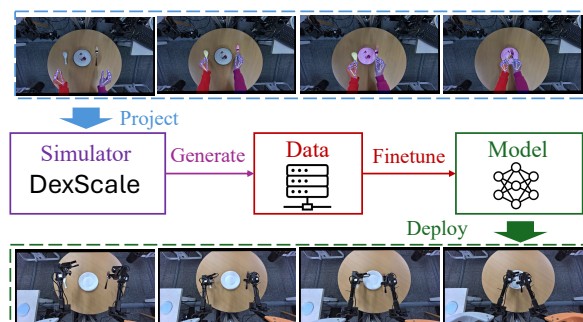

Figure 1: An example of using DexScale. Given a human video, DexScale simulates a robot skill dataset for training deployable control policy without requiring realistic data.

making processes (Paolo et al., 2024). In recent years, robots have emerged as a widely studied form of embodiment (Duan et al., 2022). In principle, robot control skills can be highly versatile—generalizing across various domains and incorporating multiple forms of sensory data. Prioritizing the availability of sensors and the impact of real-world application, we primarily focus on advancing visuomotor robotic learning for dexterous manipulation.

Many recent studies have proposed learning-based systems for manipulating various object based on image sensor signals. For example, modular approaches first predict affordances, such as grasp poses (Mousavian et al., 2019; Sundermeyer et al., 2021; Wu et al., 2020; Jiang et al., 2021; Fang et al., 2023) or keypoints (Manuelli et al., 2019; Qin et al., 2020; Mo et al., 2021; Yuan et al., 2024), for different tasks and train a low-level policies to complete these task. On the other hand, end-to-end approaches aim at learning a policy model that directly maps visual inputs to robot actions methods (Levine et al., 2016; Zhang et al., 2018; Florence et al., 2020; Chi et al., 2023). While these methods have promising performance in specific scenarios, the learned skills lack generalizability to diverse environments, making them unsuitable for deploying in the wild.

With advancements in learning algorithms, large-scale datasets, and hardware systems, large foundation models have achieved human-level performance across a wide variety of environments and downstream tasks. Their success inspired the construction of foundation models for robot control such as RDT (Liu et al., 2024b), Open-VLA (Kim

et al., 2024), ACT (Zhao et al., 2023), and Octo (Team et al., 2024). Training these models depends on massive amounts of data. Apart from the text and image data readily available on the internet, learning robotic control policies requires multi-modal sensory signals, robot self-states (i.e., proprioception), action trajectories for end-effectors, and body joints, all situated in a calibrated 3D space with real-world physics. It often requires structured pipelines or specially designed equipment to generate such datasets.

Although previous works have developed real-world data collection systems for robotic action trajectories (Zhao et al., 2023), these systems often rely on costly equipment and substantial manpower. Furthermore, in the absence of standardized data collection protocols, significant effort is required to align datasets collected from different systems. To overcome these limitations, an alternative method is data synthesis based on the simulated robot learning environment. In principle, simulated data generation can produce an infinite amount of robot skills (Wang et al., 2024), significantly addressing the "data hunger" challenge in training large-scale robotic control policy. However, skills trained on simulation data face significant challenges when deployed to real-world environments due to: *1) Semantic Mismatch*: There is no guarantee that the semantics of the simulated environment align with the target environment. A policy trained in a misaligned simulation is unlikely to be successfully deployed in practice. *2) Simulation Discrepancy*: Despite substantial advancements in simulators with physics-realistic and photo-realistic rendering (Todorov et al., 2012; Xiang et al., 2020; Makoviychuk et al., 2021), discrepancies between simulated and real-world environments are inevitable, continuing to hinder deployment performance.

In this paper, we introduce a data engine, DexScale, to address these challenges and facilitate the training of Sim2Real transferable dexterous manipulation policies based on visual observations. Figure 1 illustrates an example application of DexScale. Given task-descriptive observations (e.g., human demonstration), DexScale generates a simulated dataset, based on which the robot control policy can generalize to realistic tasks in a zero-shot manner.

To ensure semantic alignment, DexScale introduces a data projection mechanism that enables practitioners to incorporate heterogeneous formats of environmental and action priors from the target tasks into the simulator. These priors capture both static features (e.g., object types and layouts) and dynamic features (e.g., human demonstrations and interactions with objects), enabling the simulator to generate desired environments for learning preferable skills in an effective and reliable manner.

To address the Sim2Real discrepancy, DexScale employs a strategic data scaling approach, incorporating an automatic domain randomization process to generate diverse learning

configurations. This enables DexScale to actively discover effective robot control trajectories across a wide range of scenarios. Additionally, to remove task-irrelevant features from the dataset, DexScale adapts visual sensory data from both real and virtual environments into object-oriented and pose-affordance representations. The resulting trajectory dataset facilitates the development of imitation policies that can be seamlessly deployed in real-world applications.

We demonstrate that the DexScale pipeline can be seamlessly integrated into the training of widely studied embodied robot policies, including Acting Transformer (Zhao et al., 2023), Diffusion Policy (Chi et al., 2023), and Robot Diffusion Transformer (Liu et al., 2024b). By leveraging DexScale, these methods achieve zero-shot deployment for desired tasks. Such deployments can be effectively scaled across various robot configurations (e.g., single-arm and dual-arm setups with different robot models) and diverse manipulation tasks, including object grasping, articulated object (e.g., box) manipulation, and table rearrangement. With extensive performance evaluation and case studies, we demonstrate that DexScale can significantly advance the Sim2Real learning of deployable policies across multiple scenarios.

## 2. Related works

**Large-Scale Dataset for Robotic Learning.** Building large-scale datasets is essential to training generalizable robotic policies. For instance, to support the development of adaptable robotic controllers, RoboNet (Dasari et al., 2019) and D4RL datasets (Fu et al., 2020) capture rich robot actions across different manipulation tasks. In recent years, to enhance the training of end-to-end foundation models, extensive data collection systems have been built across different scenarios, including object manipulation (e.g., Bridgedata (Ebert et al., 2022; Walke et al., 2023)) and language conditioned and vision-based tasks (e.g., RT-1 (Brohan et al., 2023) and RT-2 (Zitkovich et al., 2023)). To address the scarcity of robot training data, MimicGen (Mandlekar et al., 2023) and DexMimicGen (Jiang et al., 2025) leverage scaled human teleoperation motions to generate robotic manipulation data. To further improve generalization performance, RT-X (O'Neill et al., 2024) introduced the Open X-Embodiment (OXE) datasets, which are cross-domain and were collected from multiple types of robots across numerous institutions. A fundamental challenge in constructing real-world datasets lies in the high costs associated with equipment and manpower.

**Environment Simulation for Robotic Learning.** Instead of collecting data from real-world, an alternative method is generating robot skills (e.g., action trajectories) from a simulated environment based on physics engines, such as MuJoCo (Todorov et al., 2012), PyBullet (Coumans & Bai,

2016–2021), Isaac Gym (Makoviychuk et al., 2021) and Sapien (Xiang et al., 2020). In this context, numerous virtual environments have been proposed for a variety of tasks, including dexterous manipulation (e.g., RLBench (James et al., 2020), ManiSkill (Tao et al., 2024), and RoboCasa (Nasiriany et al., 2024)), robot skill imitation (e.g., RoboMimic (Mandlekar et al., 2021)), physical interactions (e.g., ThreeDWorld (Gan et al., 2022)), rearrangement (Batra et al., 2020), high-level reasoning (e.g., Alfred (Shridhar et al., 2020) and CabiNet (Murali et al., 2023)), and generative skill learning (e.g. GenAug (Chen et al., 2023b), RoboGen (Wang et al., 2024) and GenSim (Hua et al., 2024)). To better reflect the realistic control environment, Habitat 3.0 (Puig et al., 2024) and AI2-THOR (Kolve et al., 2017) enable embodied agents to interact with photo-realistic environments for navigation and manipulation tasks. The continuously works ProcTHOR (Deitke et al., 2022), expand AI2-THOR with procedurally generated environments to enhance generalization. These simulators aim to benchmark robotic skill learning by offering diverse environments and tasks. However, transferring the learned control skills to real-world scenarios often requires additional fine-tuning with realistic data. The development of simulators for zero-shot Sim2Real deployment remains a challenge.

**Sim2Real Generalization.** In the application of robot control, Sim2Real generalization techniques often involve: 1) *Domain randomization* tackles Out-of-Distribution (OoD) scenarios in practical applications augmenting the training dataset with randomized visual and physical features (Chen et al., 2022a). To determine the scale of randomization, recent studies consider updating the distribution of randomization parameters by automatic learning (OpenAI et al., 2019), active exploration (Mehta et al., 2019), Bayesian update (Muratore et al., 2021b;a), offline inference (Tiboni et al., 2023) and continual learning (Josifovski et al., 2024). 2) *Domain adaption* addresses the gap between simulated and real-world domains by mapping them into a feature space. Commonly studied spaces include depth images (Agarwal et al., 2022; Cheng et al., 2024), point clouds (Lobos-Tsunekawa & Harada, 2020; Qin et al., 2022; Chen et al., 2023a; Hua et al., 2024), and environmental dynamics (Memmel et al., 2024). 3) *Feature alignment* is essential when the learned policy depends on privileged information that is inaccessible in real-world applications. A commonly studied approach is knowledge distillation, where skills are transferred from the learned policy to a deployable policy using a teacher-student framework (Kumar et al., 2021; Qi et al., 2022; Agarwal et al., 2022; Cheng et al., 2024).

## 3. Problem Formulation

**Simulation Environments for Robotic Learning.** The goal of DexScale is to learn robot control skills based on simu-

lated environments. These environments can be modeled as an episodic Markov Decision Process (MDP), represented as $\mathcal{M} = (\mathcal{S}, \mathcal{A}, \mathcal{P}_\mathcal{T}, R, \gamma, \rho_0)$, where:

- a state $s_t \in \mathcal{S}$ captures the semantic information of a scene, encompassing the configuration (e.g., layouts, appearance, and physical characteristics) of various types of objects and the robots subject to control.

- an action $a_t \in \mathcal{A}$ define control signals for robot. The common representations of action include the end effector pose of the robot, the angle of each joint, the torque applied to each joint, and the velocity of each joint.

- Transition function $\mathcal{P}_\mathcal{T}$ characterizes the impact of robot action $a_t$ to the configuration of current state $s_t$, thereby projecting the $s_t$ to a new scene represented by $s_{t+1}$ .

- Reward function $R(s, a)$ captures various degrees of optimality of the robotic agent in completing targeted tasks after acting $a$ in state $s$. For instance, $R(s, a)$ can assess "hard" optimality (i.e., sparse rewards), which determines whether a task is completed (e.g., whether the target object is successfully grasped) or "soft" optimality (i.e., dense rewards), which measures the degree of task completion (e.g., how close the gripper is to the target object).

- $\rho_0$ denotes the initial state distribution. Our DexScale support projecting the initial state from realistic scene or automatically generating a new scene.

- $\gamma \in (0, 1]$ denotes the discounting factor which weights the importance of future rewards relative to immediate rewards in decision-making processes such that a lower $\gamma$ emphasizes immediate rewards more heavily, and a higher $\gamma$ gives greater significance to future rewards.

Note that we consider an episodic MDP with stationary policy $\pi(a|s)$, which depends only on the current state and is invariant to the decision time step $t$. Additionally, in many environments, robot sensors only observe partial information about the underlying states. In this case, a common method is to embed the historical observations into the state, such that $s_t = [o_t, o_{t-1}, \dots]$, where $o_t$ represents the partial observation at time step $t$. As a data engine, DexScale supports the automatic discovery of *skills*, denoting the sequence of *action trajectory for finishing a task* $\tau = [s_0, a_0, \dots, s_T, a_T]$. To acquire these skills, DexScale automatically decomposes long-term tasks (e.g., cooking a meal) into atomic tasks (e.g., chopping potatoes). For tasks with a "hard" optimality (e.g., whether the target object is successfully grasped), DexScale primarily relies on motion planning techniques (e.g., antipodal point detection (Chen & Burdick, 1993) combined with inverse kinematics (D'Souza et al., 2001)). For tasks with "soft" optimality (e.g., how close the gripper is to the target object), DexScale supports the automatic design of dense reward functions (Ma et al., 2024) and employs Reinforcement Learning (RL) algorithms (Sutton, 2018) to acquire control skills.

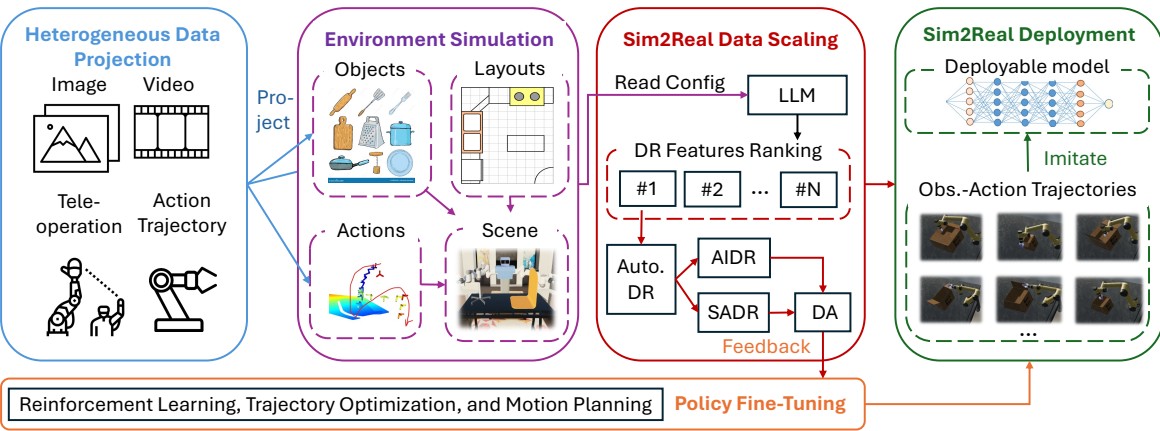

Figure 2: The pipeline of DexScale operates as a data engine, taking task-descriptive data as input and generating a skill dataset to support Sim2Real transfer. This enables the zero-shot deployment of robot policies in realistic environments. The details of these components are introduced in Section 4.

**Sim2Real Domain gap.** The skills learned in a simulated environment often face a significant domain gap when applied to the deployment environment (Xie et al., 2024). In this section, we systematically analyze the factors that cause the gap in deploying models trained with simulated data onto real robots. As it is illustrated in Table 1, we analyze by inspecting the steps in a simulation-perception-acting loop. Such analyses would give us insights on how to better design the simulated robotic learning environment that can effectively close the Sim2Real gap.

*Simulation Stage.* A major factor impacting Sim2Real performance is the inaccuracy in simulating real-world scenarios. Notably, the semantic information in the simulated environment often does not align with that of the realistic environment. For instance, objects may be inaccurately scaled, either oversized or undersized, and the location and shapes may not accurately reflect those found in the real world. More importantly, there is no guarantee that the simulated environment can fully cover the realistic semantic distributions, and the disparity between simulated and real-world environments can result in Out-of-Distribution (OoD) scenarios during deployment, making the planned motions less adaptable to the deployment environment.

*Perception Stage.* A primary application of a simulator is to replicate or reconstruct perception from realistic sensors. A critical factor affecting Sim2Real performance during this process is the inconsistency in hardware setups between the simulated and real environments (e.g., differences in the position, orientation, and field of view of cameras).

*Acting Stage.* Embodied agents control various types of embodiments (e.g., arm, and hand) by executing robot actions. In this stage, mismatches between embodiments can impact deployment. For example, using different robots during simulation and deployment, or having URDF configuration errors, can lead to discrepancies. More importantly,

the physical properties in the realistic environment might misalign with those in the simulated environment, inducing inconsistent outcomes of implementing the same action.

Table 1: Features that might induce Domain Gap. The determination of such parameters is a recent study of the generalization gap (Xie et al., 2024).

| | |
|---|---|
| Simulation Stage | Lighting, Table Texture, Background, Distractors, Object Locations, Object Orientations, Object Texture and Object Shape |
| Perception Stage | Camera Position, Camera Orientation, Camera Field of View |
| Acting Stage | Robot Configuration, Physics Properties |

## 4. Data Engine for Sim-to-Real Generalization

To acquire robot control skills that effectively bridge the aforementioned Sim2Real gap, DexScale can serve as an automated data engine. As shown in Figure 2, DexScale takes descriptive data (e.g., scenes and demonstrations) from the target task as input and generates a scalable dataset that supports the efficient training of Sim2Real-deployable policies. We detail the design of DexScale in the following.

### 4.1. Heterogeneous Data Projection

The goal of heterogeneous data projection is to map descriptive observations of the target task from real-world applications to a simulated environment, ensuring the applicability of the generated skills. To achieve this, DexScale primarily supports two key types of projection: scene projection and action trajectory projection, as detailed below.

**Scene Projection.** To overcome the Sim2Real gap, a critical challenge to handle the semantics disparity between simulated and real-world environments. For example, in the

scene constructed by generative simulation, there is no guarantee the distribution of objects and their configurations can semantically align with these in realistic scenarios. To overcome this limitation, DexScale enables projecting the *static* scene information to the simulated environment. Specifically, inspired by Digital Cousin (Dai et al., 2024), DexScale extracts relevant per-object information from the input RGB image. Using this information, DexScale matches each detected object to their "digital cousins" (i.e., similar objects) in the asset dataset. For the articulated objects (e.g., drawers and boxes), DexScale can post-process them to create a fully interactive simulated scene by matching them with CAD models and the generated objects (Liu et al., 2024a). Besides the automatic scene project, DexScale provides user-friendly interfaces to support manual adjustment of the retrieved objects and the generated scenes. This adjustment is essential if the single-view image can capture only partial information about the scene (e.g., details on sides not visible to the camera are unknown) or if the retrieved scene can not match the realistic semantics.

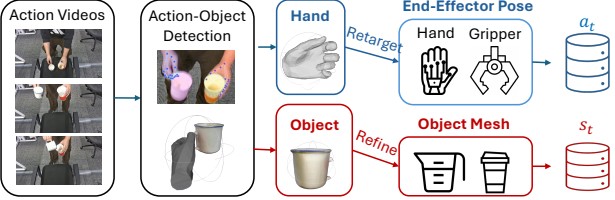

Figure 3: The flowchart of action-trajectory projection.

**Action-Trajectory Projection.** Given an egocentric video of humans manipulating objects (e.g., picking up a teapot and pouring tea into a paper cup), DexScale projects actions and objects from realistic human motion onto robot control signals within the simulated environment. We illustrate this process in Figure 3. This action trajectory captures *dynamic* information from human motion and can act as the seeding trajectories for the following skill scaling.

1) *Robot-Action Projection.* DexScale enables detecting movements of hand models from video data and retargeting these models to robotic end effectors, such as grippers and dexterous hands, thereby transforming human hand poses into those compatible with robotic devices. Within thin process, a significant challenge is the disparity in the Degrees of Freedom (DoF) between the hand model and the robotic end effectors. For instance, a robot hand might possess 6 DoFs, whereas the human hand model features over 20 DoFs. To align the movements of the human and robotic hands effectively, it is essential to consolidate multiple DoFs from the human hand into a single DoF of the robotic hand, ensuring minimal impact on model performance. In the case of retargeting to a gripper, that has only one DoF (either open or closed), we select two fingers to act as pivotal points for picking up and placing objects, and then retargeting these fingers' movements to the jaw movements of the gripper.

2) *Robot-Object Interaction.* A crucial prerequisite for action projection is reconstructing the objects and their relationships with the operators (Liu et al., 2024c). For example, as shown in Figure 3, it is essential to determine the 3D pose at which the end effectors (e.g., hands) interact with the objects (e.g., the cup), thereby ensuring the projected action trajectory can accurately reflect the interaction under real-world physics. To achieve this goal, DexScale reconstructs the 3D meshes and the poses of objects from the sequences of images in the action trajectory video data. The sequence of object poses, along with the object models, is then jointly optimized with the poses of the target end effectors captured in the robot-action projections (as mentioned above), until they are properly aligned. This process is fully automatic without any manual intervention. For each image, the goal of joint optimization is to refine the interaction between objects and end effectors, accurately capturing realistic contact dynamics such as grasping forces and stability. DexScale further refines the trajectory by ensuring smoothness and plausibility for the sequence of actions. Figure 6 shows an example of the recovered 3D mesh.

### 4.2. Environment Simulation for Robot Learning

DexScale enables the conditional generation of a robot learning environment based on the projected data. This environment can simulate static and dynamic features by integrating scenes and action trajectories as described below.

**Scene Simulation.** DexScale supports the construction of different scenarios by leveraging scene projection, which focuses on providing static features such as shape, pose, appearance, object layout, and other background details. However, since our scene projection primarily relies on single-view or multi-view images, there is no guarantee that the scene is fully observable, nor that these visual observations capture complete information about the scene. As a result, in addition to the projected information, DexScale enables the automatic construction of a complete scene based on the available data. To achieve this, we follow (Wang et al., 2024; Hua et al., 2024) and leverage large foundation models, such as GPT-4, to generate the scene configuration based on the projected information and task description. DexScale supports retrieving objects in the scene from the Objaverse-XL dataset (Deitke et al., 2023) or generating them based on language descriptions. Furthermore, if the generated scene does not align with practical requirements, DexScale provides a user-friendly interface that allows users to manually adjust and refine the generated scenarios.

**Action-Trajectory Simulation.** The objective of action simulation is to generate continuous action trajectories for the robot, enabling it to execute the tasks effectively and achieve the desired results. To construct such action trajectories, DexScale primarily relies on the projected poses of end effector extracted from human video, or the robot

joint positions collected from tele-operation. For the projected end-effector poses, DexScale utilizes a generalized Inverse Kinematics (IK) algorithm to compute the corresponding joint configurations, ensuring precise achievement of the desired end-effector pose. To maintain a smooth trajectory, joint-space interpolation is applied across the motion. In scenarios requiring obstacle avoidance, DexScale integrates widely used motion planning algorithms, such as RRT-Connect (Kuffner & LaValle, 2000), to efficiently generate collision-free trajectory. In applications where action projection is unavailable, DexScale supports the automatic design of reward and goal functions by consulting with large language models (Ma et al., 2024). This enables the RL and trajectory optimization algorithms to effectively learn robot control skills. To ensure that the simulated actions align with the projected action trajectories, DexSca le allows for the replay and automatic refinement of these trajectories via learning-based methods. Using these refined trajectories, we can train an imitation policy $\pi_{\theta^0}(a|s)$, for robot control. However, directly deploying $\pi_{\theta^0}(a|s)$ in real-world applications is challenging due to the existing Sim2Real gap between the simulated and real-world environments. To address this issue, DexScale incorporates the following Sim2Real scaling techniques to bridge this gap.

### 4.3. Sim2Real Data Scaling

In a simulated environment, various factors can contribute to the Sim2Real gap (Section 3), undermining the effectiveness of generalizing the learned robot control policy to real-world applications. To address this gap, DexScale primarily focuses on establishing a pipeline for automatic domain randomization and adaptation, as outlined below.

**Automatic Domain Randomization.** Domain Randomization (DR) enables Sim2Real transfer by varying simulated environments (Chen et al., 2022b). We achieve DR by modifying the original environment $\mathcal{M}$ with configuration $\xi$, creating a new environment $\mathcal{M}^{\xi}$. In robotic learning environments, to better assess the effects of different DRs, we categorize these DRs based on their impact on agents' actions in the following.

1) *Action-Invariant Domain Randomization (AI-DR).* Within our DexScale, the goal of AI-DR is to prevent the embodied agent from overfitting to the simulated features that are irrelevant or ineffective in completing a task (e.g., the lighting condition on manipulations). By removing the effects of these features and concentrating only on essential features for finishing a task, the agent experiences a less complex sim-to-real gap, and the learned skills have stronger transferability. For these DRs (configured by $\xi$), they have limited impact on the agents' decisions and movements. To better formulate this consistency, we assume the optimal actions remain uninfluenced before and after applying AI-DR to an environment $\mathcal{M}$, so that

$\pi_{\theta\xi}^*(a|s) = \pi_{\theta^0}(a|s) \; \forall (s,a) \in \mathcal{S} \times \mathcal{A}$ where $\pi_{\theta\xi}^*$ denotes the optimal policy under the DR environment $\mathcal{M}^{\xi}$.

2) *Semantic-Aware Domain Randomization (SA-DR).* The goal of SA-DR is to generalize the embodied agents' skills from the source environment $\mathcal{M}$ to different variations of DR environment $\mathcal{M}^{\xi'}$. This approach allows the agent's skills to overcome the Sim2Real gap, provided that the gap falls within the range of simulated variations. To better handle these DRs (configured by $\xi'$), the agent must adapt the decisions and movements so that the optimal actions may vary before and after applying SA-DR to an environment $\mathcal{M}$. The adapted policy $\pi^{*,\xi'}$ can be represented as:

$$\pi_{\theta\xi'}^* = \arg\max_{\pi_{\theta\xi'}} \mathcal{J}(\mathcal{M}^{\xi'}, \pi_{\theta\xi'}) - \text{Div}(\pi_{\theta\xi'} \| \pi_{\theta^0}) \quad (1)$$

Where $\mathcal{J}(\mathcal{M}^{\xi'}, \cdot)$ denotes the optimality function under the environment $\mathcal{M}^{\xi'}$ and $\text{Div}(\pi_{\theta\xi'} \| \pi_{\theta^0}^*)$ indicate the divergence between policies $\pi_{\theta\xi'}$ and $\pi_{\theta^0}^*$. For example, in the continual learning setting, by setting $\text{Div}(\pi_{\theta\xi'} \| \pi_{\theta^0}^*) = \lambda \|\theta^{\xi'} - \theta^0\|$, the objective (1) can effectively represent the Elastic Weight Consolidation (EWC) objective. More importantly, as opposed to the action invariance in AI-DR, we must finetune the policy $\pi_{\theta^0}^*$ based on the updated objective $\mathcal{J}(\mathcal{M}^{\xi'}, \pi_{\theta\xi'})$ under the new environment $\mathcal{M}^{\xi'}$ after applying SA-DR. In this setting, both RL and motion planning algorithms can be utilized to fine-tune the original policy and adapt it to the new environment.

*Automating DR.* Each DR configuration $\xi$ corresponds to a control policy $\pi_{\theta\xi}^*$, which captures skill for solving the task under the DR environment $\mathcal{M}^{\xi}$. By generating skills $\tau$ using $\pi_{\theta\xi}^*$, we can construct a Sim2Real dataset $\mathcal{D}_{\text{DR}}$. Based on this dataset, the imitation policy can effectively bridge the Sim2Real gap, provided that the experimented DR configurations and their combinations accurately capture the underlying Sim2Real discrepancies. To achieve this goal, a critical prerequisite is being able to characterize the types, ranges, and styles of applied DRs. Within DexScale, our DR selection primarily focuses on the key factors for the Sim2Real gap (Section 3). For example, given an environment $\mathcal{M}$, we must first identify which features are suitable for DR and determine the corresponding types of DRs, such as AI-DR and SA-DR. Additionally, for a specific DR parameter $\xi$, we need to model the distribution of DR parameters, $p_\phi(\xi) \in \Delta^\Xi$ (Chen et al., 2022a). While traditional simulators often manually specify $p_\phi(\xi)$, DexScale seeks to automate the selection of DR features and modeling $p_\phi(\xi)$ conditioning on specific tasks. As illustrated in Figure 2, DexScale integrates a large foundation model into the DR process by leveraging its ability to rank DR features based on the configuration of the simulated environment. For each DR configuration $\xi$, DexScale employs the ADR algorithm (OpenAI et al., 2019) to calculate and update $p_\phi(\xi)$ by incorporating feedback from the fine-tuned policies and their outcomes in the environment. Domain

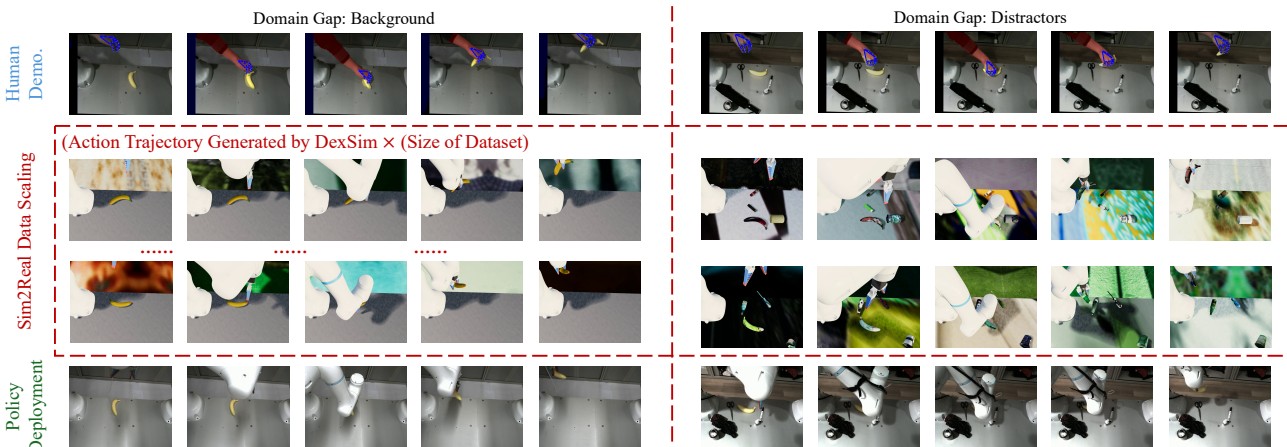

Figure 4: Visualizing the input demonstration (top row), examples of the scaled data (middle rows), and the realistic deployment of the robot policy (bottom row). The Sim2Real domain gaps are background (left) and distractors (right).

randomization alters the size, shape, pose, and texture of objects during training, enabling the model to generalize its learned skills to a wide variety of objects. As a result, the model can effectively handle real-world objects, even if they differ significantly from those seen in simulation.

**Domain Adaption (DA).** To more effectively bridge the gap between simulation and reality, domain adaptation project the observations from the simulated and realistic environment to a into a unified target space. DexScale supports various types of DAs, including 1) *Object-Oriented Representations*, where DexScale excludes background information from image observations from RGB cameras and focuses only on each object. To better capture the geometric and spatial properties of objects, DexScale can map objects into point clouds based on their masks within the simulated environment. 2) *Pose Affordance Representations*, where DexScale selects key affordances (e.g., pose) from an action trajectory, capturing the critical poses necessary to complete the task. For example, when grasping an object, the affordance represents the gripper's pose when it first contacts the object, enabling the robot to complete the task by reaching these poses using inverse kinematics. These prediction targets, derived from affordance representations, occur less frequently than full action trajectories, thereby reducing accumulated errors during deployment. With these methods, DexScale can map $\mathcal{D}_{DR}$ to the post-adaptation dataset $\mathcal{D}_{DR+AR}$.

### 4.4. Sim2Real Depolyment

DexScale adopts a data-driven approach to achieve Sim2Real deployment. Specifically, the deployable policy is trained by imitating the data trajectories in $\mathcal{D}_{DR+AR}$. Since $\mathcal{D}_{DR+AR}$ captures diverse and rich action trajectories across various environments, the resulting control policy remains effective under the Sim2Real gap, provided the realistic environment lies within the support of the dataset's distribution.

In the following experiment section, we demonstrate that various high-performing imitation models—including action transformer policies (Zhao et al., 2023), diffusion policies (Chi et al., 2023), and Vision-Language-Action (VLA) policies (Liu et al., 2024b)—can be trained and deployed to control real-world robots under different environments.

## 5. Experiments

The pipeline of DexScale is generic and agnostic to simulation platforms (Appendix A.1 shows our specifications). We evaluate the performance of Sim2Real deployment across various applications and assess the validity of Real2Sim projection in terms of action and object mapping by addressing the following questions: 1) **Generalizability:** How effectively does DexScale bridge the Sim2Real gap between simulated environments and real-world applications? 2) **Scalability:** Can the control skills learned by DexScale be scaled across different models and embodiments?

### 5.1. Generalizability: Bridging the Sim2Real Gap

**Experiment Setting.** This experiment aims to quantify DexScale by how effectively it can overcome the Sim2Real gap. We first simulate a baseline environment by projecting the real-world scene into a simulation (via our Real2Sim projection in Section 4.1) and then manually refining it to ensure alignment with the realistic setting. Using this baseline environment, we can modify its parameters and intentionally introduce the Sim2Real gaps that frequently occur in practice based on our analysis (see Table 1). We summarize such modifications in Appendix A.2. DexScale is then evaluated based on its effectiveness in overcoming these gaps and acquiring robust skills that can be seamlessly applied to real-world applications. To gain a deeper understanding of DexScale, we conducted an ablation study by removing either the strategic Domain Adaptation (DA) or Domain Randomization (DR) components from our DexScale dataset. This resulted in a skill only dataset that records

Table 2: Success rates of imitation policies learned by different datasets under various Sim2Real gaps. For the first eight domain gaps, we employ the transformer-based policy (Zhao et al., 2023) to tackle grasping tasks. For the last two domain gaps, we use the diffusion-based policy (Chi et al., 2023) to address the open-box task.

| Dataset | Skills Only | | Skills+DR | | Skills+DA | | DexScale | |
|---|---|---|---|---|---|---|---|---|
| Domain Gap | Sim. | Real. | Sim. | Real. | Sim. | Real. | Sim. | Real. |
| Light | 64/100 | 0/10 | 75/100 | 4/10 | 74/100 | 1/10 | 73/10 | 4/10 |
| Object Texture | 49/100 | 1/10 | 81/100 | 5/10 | 75/100 | 3/10 | 83/100 | 6/10 |
| Table Texture | 55/100 | 2/10 | 81/100 | 4/10 | 73/100 | 2/10 | 82/100 | 4/10 |
| Background | 67/100 | 1/10 | 83/100 | 3/10 | 71/100 | 1/10 | 82/100 | 3/10 |
| Distractors | 46/100 | 0/10 | 65/100 | 2/10 | 73/100 | 2/10 | 72/100 | 4/10 |
| Camera Position | 72/100 | 2/10 | 73/100 | 4/10 | 71/100 | 2/10 | 80/100 | 5/10 |
| Camera Orientation | 69/100 | 0/10 | 74/100 | 4/10 | 66/100 | 0/10 | 78/100 | 3/10 |
| Camera Field of View | 69/100 | 1/10 | 80/100 | 3/10 | 66/100 | 1/10 | 82/100 | 6/10 |
| Object Pose | 15/100 | 1/10 | 33/100 | 2/10 | 42/100 | 3/10 | 69/100 | 4/10 |
| Object Shape | 11/100 | 0/10 | 27/100 | 2/10 | 38/100 | 2/10 | 61/100 | 3/10 |

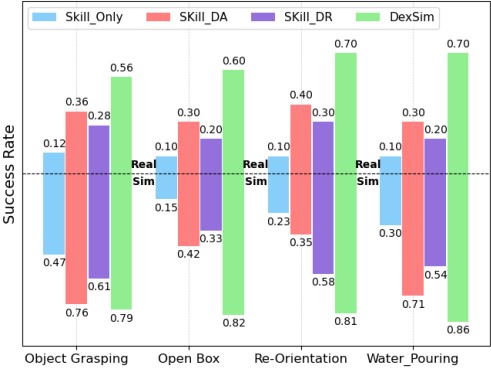

Table 3: Robot control performance for different tasks under both realistic (upper) and simulated (lower) environments.

only the action trajectory for the target task in the simulated environment. Additionally, To prevent overfitting, we augment the skill dataset by adding random noise to skills.

To evaluate the model's control performance in bridging the Sim2Real gap, we trained the control model using different methods and assessed its performance through 100 trials in the simulated environment and 10 trials in the real-world environment. The success rates are presented in Table 2. We found that the Sim2Real data scaling performed by DexScale significantly enhances the control model's ability to consistently bridge the domain gap across both simulated and real-world environments. Removing the designed DA and DR components leads to a notable decline in performance. To better understand the performance of DexScale, Figure 4 shows keyframes of scaled data and policy deployment in realistic scenarios (Appendix B.2 shows more examples). To address the domain gap caused by distracting items (e.g., umbrellas, scissors, or pens not being the target in the right columns of Figure 4), an interesting observation is that DexScale not only randomizes different objects but also backgrounds and textures in the scene. While textures are not directly related to distractors, they are effectively in bridging the Sim2Real gap, and DexScale discovers this relation. To better understand the difficulty of real-world performance, we include a failure case study on the project page, highlighting the following common issues: 1) unobservable grasping orientation, 2) incorrectly predicted grasp pose, 3) inaccurate grasp depth prediction, and 4) cross-chunk jitter.

**Comparison with Handcrafted DR**. In addition, we investigate an alternative approach that relies on human expertise to manually select the type and scale of DR, instead of using our DexScale. Following the empirical analysis in (Xie et al., 2024), which ranks the importance of various DR features, we apply the top-ranked features, 1) camera orientation, 2) table texture, and 3) distractors, to bridge the Sim2Real gap. We then evaluate model performance under each setting and report the results accordingly. We observe

that the success rates of pick-and-place tasks in simulation and the real world are 0.62/0.10, 0.63/0.30, 0.73/0.40, and 0.79/0.56, respectively. This increasing trend highlights the critical role of automatic DR in improving Sim2Real transfer performance.

## 5.2. Scalability across Diverse Tasks and Embodiments

**Experiment Setting.** A critical prerequisite for developing a Sim2Real simulator is its ability to scale across diverse robot control tasks and environments. Unlike most prior work, which primarily demonstrates realistic performance with a single embodied robot, our approach emphasizes cross-embodiment evaluation to comprehensively assess Sim2Real performance. In this experiment, we evaluate our approach on four challenging tasks: 1) *object grasping*, which requires the robot to accurately detect objects and predict appropriate grasp poses; 2) *paper box manipulation*, involving precise control and planning to sequentially open all four flaps of a box; 3) *dual-arm table rearrangement*, where the robot must reorient both a fork and a spoon to face the front of the plate and place them accurately around it; and 4) *bottled water pouring*, which involves grasping and reorienting a water bottle to pour water precisely into a paper cup. For these tasks, we train imitation models using various architectures, including the transformer-based policy (Zhao et al., 2023) for grasping, the diffusion policy (Chi et al., 2023) for manipulation, and Vision-Language-Action (VLA) models (Liu et al., 2024b) for table rearrangement. Appendix A.3 reports the specific training configurations.

Figures 4, and 5 illustrate examples of action trajectories for the tasks of object grasping, box manipulation, and table rearrangement. To demonstrate the scalability of DexScale, the control policies are deployed on different robots, including two single-arm robots and a dual-arm robot equipped with wrist-mounted cameras. Figure 3 presents the end-to-end performance across various tasks, showcasing both real-to-sim (from task-descriptive data to the simulator) and sim-to-real (from simulation to real-world deployment) tran-

sitions. By leveraging the scaled data generated by DexScale, our models consistently outperform baseline methods across different tasks, model architectures, and robot platforms, highlighting the scalability and effectiveness of DexScale.

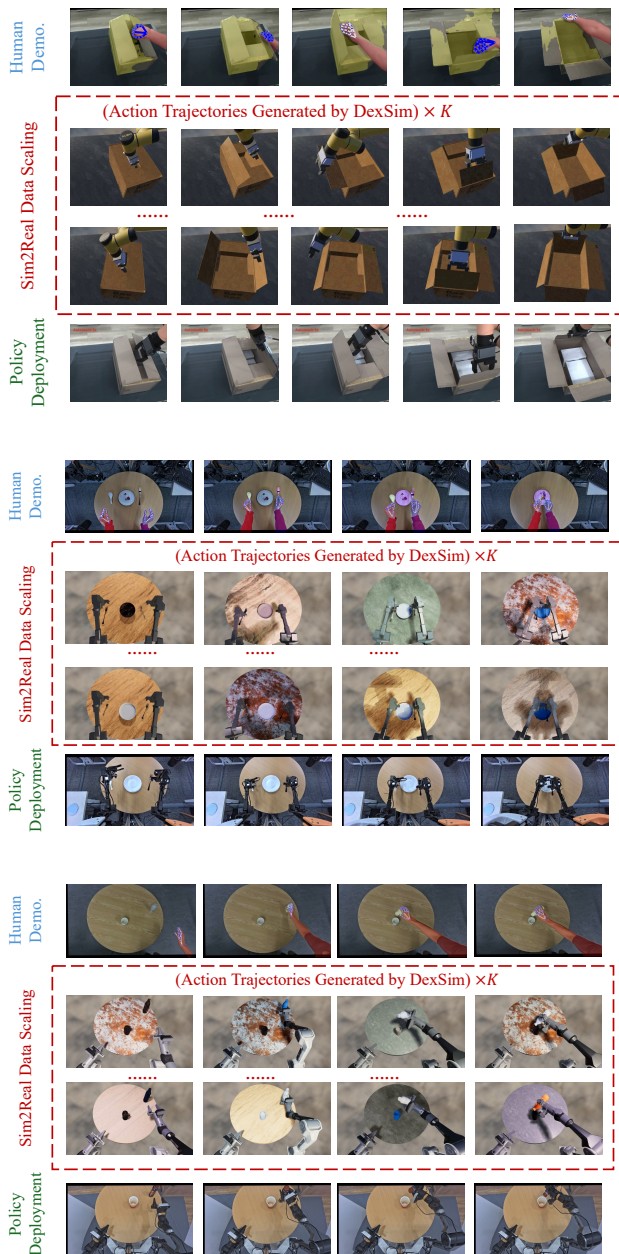

Figure 5: Visualizing the human demonstration, scaled trajectories and realistic robot action for the task of opening a box, table rearrangement and pouring water.

## 6. Conclusion

We introduce DexScale, a data engine designed to generate simulated robot action trajectories for learning deployable control policies by mapping realistic observations into simulation and utilizing automated domain randomization and adaptation for data scaling. These capabilities can be efficiently expanded to accommodate a wide range of tasks, learning models, and robotic platforms. In future work, we aim to extend DexScale to more complex robots (e.g., humanoid robots) and long-term tasks (e.g., cooking a meal).

## Acknowledgments

This work is supported in part by Science and Technology Major Program under grant KJZD20240903104008012, Guangdong-Shenzhen Joint Research Fund under grant 2023A1515110617, Guangdong Basic and Applied Basic Research Foundation under grant 2024A1515012103, and Guangdong Provincial Key Laboratory of Mathematical Foundations for Artificial Intelligence (2023B1212010001).

## Impact Statement

The broader impact of this work lies in its potential to significantly advance the field of robotics by addressing one of its most pressing challenges: the gap between simulated and real-world environments.

From an ethical perspective, DexScale promotes more inclusive and equitable innovation by making the development of robotic systems more accessible, even to those with limited resources. However, as with any technology that advances automation, there are potential societal consequences to consider. The widespread adoption of robots in tasks such as manufacturing, service, or even domestic work could lead to significant shifts in labor markets, raising concerns about job displacement and economic inequality.

In the long term, DexScale's ability to scale robotic intelligence across diverse applications could have transformative societal implications. Robots trained using DexScale could be deployed in healthcare, disaster response, education, and other critical areas, amplifying their positive impact on society. However, ensuring the ethical use of such robots will require careful consideration of privacy, safety, and accountability. Future research should also explore ways to mitigate potential biases in simulated datasets to ensure fairness and reliability in real-world deployments.

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

# A. Implementation Details

## A.1. Specification of DexScale

DexScale is a scalable, automated data engine designed to bridge the Sim2Real gap in robot skill learning by combining physically based rendering (PBR), physics simulation and high performance multi-threading system. Its architecture is optimized for generating large-scale, diverse, and realistic robot learning environments and datasets, while ensuring computational efficiency. Below, we detail its core specifications:

**System Architecture.** DexScale is built on a modular framework with three interconnected subsystems:

- *Physically-Based Rendering Engine*: Combines ray tracing and rasterization techniques to synthesize photorealistic visuals with dynamic lighting effects and physically accurate material properties, such as surface roughness, metallic reflectance, and albedo maps.

- *Physics Simulation Engine*: Employs a rigid-body dynamics solver with support for frictional contacts and articulation modeling (e.g., grippers, robotic arms)

- *Scene Composition*: Compose complex scenes with customizable objects, lighting, and backgrounds. Users can define object dynamics properties, such as mass, friction, and elasticity, or material properties to match real-world conditions.

DexScale's modular architecture is engine-agnostic, enabling seamless integration with arbitrary rendering or physics engines as backends. This design ensures adaptability to diverse use cases while decoupling core functionality from third-party dependencies.

**Robotic Simulation.** Our robot simulation module provides a unified framework for modeling, controlling, and training robotic systems through three core components:

- *Robot Abstract Models*: Predefines the parameterized templates for common robotic platforms, such as Manipulators ((e.g., 6-DoF industrial arm), Dexterous Hands and Humanoids with unified URDF description and configurable kinematic chains, inertial properties, Joint limits, etc. It also supports the customization of robot morphologies through a modular assembly of actuators and sensors.

- *Control System*: Provides forward/inverse kinematics solvers with singularity handling and motion generation capacities with trajectory interpolation and collision-aware planners. It has user-friendly interface to access robot proprioception and execute the control signal in both joint space and task space (eg, euclidean space)

- *Robot Learning Environment*: Adopts OpenAI gym-compatible API with standardized method (e.g., **reset**, **step**), and support domain randomization for both dynamics and visuals properties of the objects and scene. It includes task suites (e.g., manipulation, locomotion) equipped with configurable callback and reward functions, enabling the implementation of customized features tailored for both imitation learning and reinforcement learning.

**Data Generation.** Our data generation module enables large-scale synthesis of labeled simulation datasets through a pipeline combining assets generation, geometric processing, and automated annotation. The architecture comprises four core subsystems:

- *Scene Construction*: Combines procedural generation, leveraging parameterized templates for randomized object placement, lighting configurations, and camera viewpoints, with rule-based scene assembly (e.g., clutter or ordered arrangements) and collision-free guarantees. It integrates 3D AIGC for asset synthesis via state-of-the-art models like TRELLIS (Xiang et al., 2024), while also supporting retrieval of 3D assets from a large-scale, multi-label database.

- *Mesh Processing*: Provides an automated pipeline to ensure 3D geometry assets are simulation-ready. This pipeline includes UV mapping, geometric processing (e.g., remeshing, hole filling, and simplification), and convex decomposition for efficient collision detection.

- *Domain Randomization*: Provides interface to adjust or change the parameters of the domain randoization factors described in section 4.3.

- *Annotation Computation*: Supports diverse annotation types, including instance/semantic segmentation masks, 6D object poses, keypoints, and physics-based ground truth such as contact vectors at interaction points. It also provides interfaces to export datasets in standard formats like COCO and HDF5.

## A.2. The Details of Sim2Real Gap

- *Light.* We adjust the lighting conditions by varying the distance between the light source and the scene of the given tasks. Specifically, a 3-meter difference is introduced between the simulated and real-world environments, resulting in a darker lighting condition in the real-world setting.

- *Object Texture.* We primarily focus on the material properties and appearance, including the color, of each object. Specifically, in the real-world environment, the objects are covered with tissues, while in the simulated environment, the color and texture are tailored to match the object type (e.g., a banana has a smooth yellow surface).

- *Table Texture.* We primarily focus on the differences in table surface patterns between the simulated and real-world environments. Specifically, in the real-world environment, the table is covered with dark paper, while in the simulated environment, a table surface in a light grey table surface is used.

- *Background Texture.* Since the camera's view primarily captures the floor where the table is placed, we focus on the differences in floor surface patterns between the simulated and real-world environments. Specifically, in the real-world environment, the floor is covered with a light grey surface, while in the simulated environment, a wooden surface is used.

- *Distractors.* The distractors consist of objects that are not intended to be grasped by the gripper. In the real-world environment, these objects include a paper cup, a folding umbrella, a pull-tab can, scissors, and a spray bottle. However, in the simulated environment, these objects are intentionally removed.

- *Camera Position.* We adjust the camera position in the simulated environment. Specifically, the camera is placed 5 cm higher and 5 cm to the right compared to the position of the camera in the real-world environment.

- *Camera Orientation.* We adjust the camera orientation in the simulated environment. The camera is rotated 5 degrees counterclockwise around both the X and Z axes in the simulated environment.

- *Camera Field of View.* We adjust the camera orientation by modifying its parameters. Specifically, the focal length is set to 1.15 times the actual focal length to achieve the desired adjustments.

- *Object Pose.* We adjust the object pose in the simulated environment by applying a rotation of up to 15 degrees and shifting their x-y positions by up to 5 centimeters.

- *Object Shape.* We rescale the width, length, and height of the object in the simulated environment, with the scaling ranging from 0.9 to 1.1 times the object's size in the realistic environment.

## A.3. Model Training Configurations

### A.3.1. OBJECT GRASPING

**Model Structure.** We adopt the HumanPlus HIT model configuration and outline the following details:

- *Model Architecture:* The model employs 2 ResNet18 backbones to process visual inputs and output visual latents. Action data is projected into action latents using an MLP. The visual and action latents are then fed into 6 BERT-style transformer decoders for further processing.

- *Number of Parameters:* The total number of parameters in the model is $30.4 \, \text{M}$.

- *Activation Functions:* GELU is used as the activation function throughout the network.

**Dataset Details.** The dataset consists of 2000 trajectories for each feature and each setting (gap, domain randomization, domain adaptation, domain randomization and adaptation), with each trajectory containing 50 steps. Each trajectory includes 21-dimensional action data (comprising 6-dim joint angles, 7-dim end-effector pose under the robot base frame, 1-dim gripper open state, and 7-dim end-effector pose under the camera frame) and two $512 \times 640$ rectified images. The dataset is

split into 99% training and 1% validation. Data preprocessing involves normalization for action data, ImageNet-statistics normalization for images, and image rectification. During training, color-jitter data augmentation is applied to the images.

**Training Hyperparameters.** The model is trained with a batch size of 12 for 200,000 epochs using the AdamW optimizer with a learning rate of $1 \times 10^{-5}$. The training objective is guided by the mean squared error (MSE) loss for actions and a 1-dimensional error for image latents.

**Training Environment.** Training was conducted on 4 NVIDIA A800 GPUs, with each policy trained for 36 hours. PyTorch version 2.0.1 was used as the deep learning framework.

### A.3.2. OPEN BOX

**Model Structure.** We follow the network architecture of EquiBot (Yang et al., 2024), with several modifications to enhance sim-to-real transfer. Our model is based on the SIM(3)-Equivariant Diffusion Policy (Yang et al., 2024), with the following details:

- *Network Architecture:* The original EquiBot processes entire scene point clouds as the observed input environment, with a sampling point number of 1024 or 2048 depending on the task type. In our modified version, we use the cropped point clouds of manipulated objects as input and always set the sampling point number to 1024.

- *Mask Detection:* The 2D mask of manipulated objects is obtained automatically in simulation using the DexScale simulator. During real-world inference, the mask is detected using vision foundation models, such as Florence2 (Xiao et al., 2024) and SAM2 (Ravi et al., 2024).

- *Prediction and Observation Horizon:* The prediction horizon is set to 72, and the observation horizon is 1. Unlike the high-frequency dynamic style used in the original EquiBot (prediction-conduction-observation), we observe once and execute all predicted actions at once.

- *Stereo Vision:* Our model uses a binocular camera to capture left and right images simultaneously. The left image branch lifts 2D pixels into 3D using a stereo matching algorithm (Xu et al., 2023). This approach significantly reduces domain gaps caused by hardware differences between real and simulated environments.

- *Number of Parameters:* The total parameter size is 111.43 MB.

**Dataset Details.** The dataset consists of 50 demonstrations in the training set, with each demonstration containing 75 timesteps. The dataset is split into 50 demonstrations for training and 10 demonstrations for testing. Additionally, 100 evaluation trials are conducted in the simulator to assess performance.

**Training Hyperparameters.** The training process uses a batch size of 16 and runs for 150,000 epochs. The model is optimized using the Adam optimizer with a cosine learning rate scheduler, starting with a learning rate of $5 \times 10^{-4}$. The training objective is guided by the Chamfer Distance loss function for point clouds and the Mean Squared Error (MSE) loss for action predictions.

**Training Environment.** Training was performed on a single NVIDIA A100 GPU for a total of 48 hours. PyTorch version 2.0.1 was used as the deep learning framework.

### A.3.3. TABLEWARE REARRANGEMENT

**Model Structure.** We follow the original Robotics Diffusion Transfromer (RDT) model configuration (Liu et al., 2024b) and make the following modifications:

- *Vision Encoder:* Replace SigLip with DINOv2 (base) with registers as the vision encoder.

- *Image Conditions:* Use one stereo camera along with two wrist-mounted RGB cameras, resulting in the following image conditions: 2 RGB images, 1 disparity image, and 2 additional RGB images.

- *Mask Predictor Module:* Add a small mask predictor module consisting of 3 convolutional layers to predict the mask for each image. The updated image condition is then computed using the following equation:

$$\text{img\_cond\_new} = \text{img\_cond} + \text{mask} \cdot \text{mask\_embed}$$

- *Transformer Block Settings:* The transformer block is configured with a hidden size of 512, a depth of 8 layers, and 16 attention heads.

- *Model Size:* The total number of parameters is $116.66$ MB.

**Dataset Details.** The dataset used in this work consists of multiple trajectories designed to capture diverse scenarios for training and evaluation. Specifically: *1) Number of Trajectories:* The dataset includes 200 trajectories, each consisting of 200 steps. *2) Data Preprocessing:* The data is preprocessed using the same pipeline as described in the original RDT setup.

**Training Hyperparameters.** The training process utilizes a batch size of 8 and runs for 40,000 iterations. The model is optimized using the Adam optimizer with a cosine learning rate scheduler, starting with a learning rate of $1 \times 10^{-4}$. The training objective is guided by the Mean Squared Error (MSE) loss function.

**Training Environment.** The training was conducted on a single NVIDIA A100 GPU using PyTorch version 2.0.1 as the deep learning framework.

### A.3.4. HARDWARE DETAILS FOR DIFFERENT TASKS

Table 4: Robot Specifications for Different Tasks

| Task | Robot Name | DOF | Maximum Reach (mm) | Maximum Payload (kg) |
|---|---|---|---|---|
| Object Grasping | Rokae SR3 | 6 | 705 | 3 |
| Open Box | AUBO I5 | 6 | 886.5 | 5 |
| Tableware Rearrangement | WidowX 250 S | 7 | 650 | 0.25 |

## B. More Experiment Results

### B.1. Human-Object Interaction Example

Figure 6 visualize the input image alongside the output hand-object interactions as a 3D mesh, which can be directly projected into a simulated environment. To enhance the visualization of the 3D mesh details, we present observations from six different angles.

### B.2. Examples of using DexScale

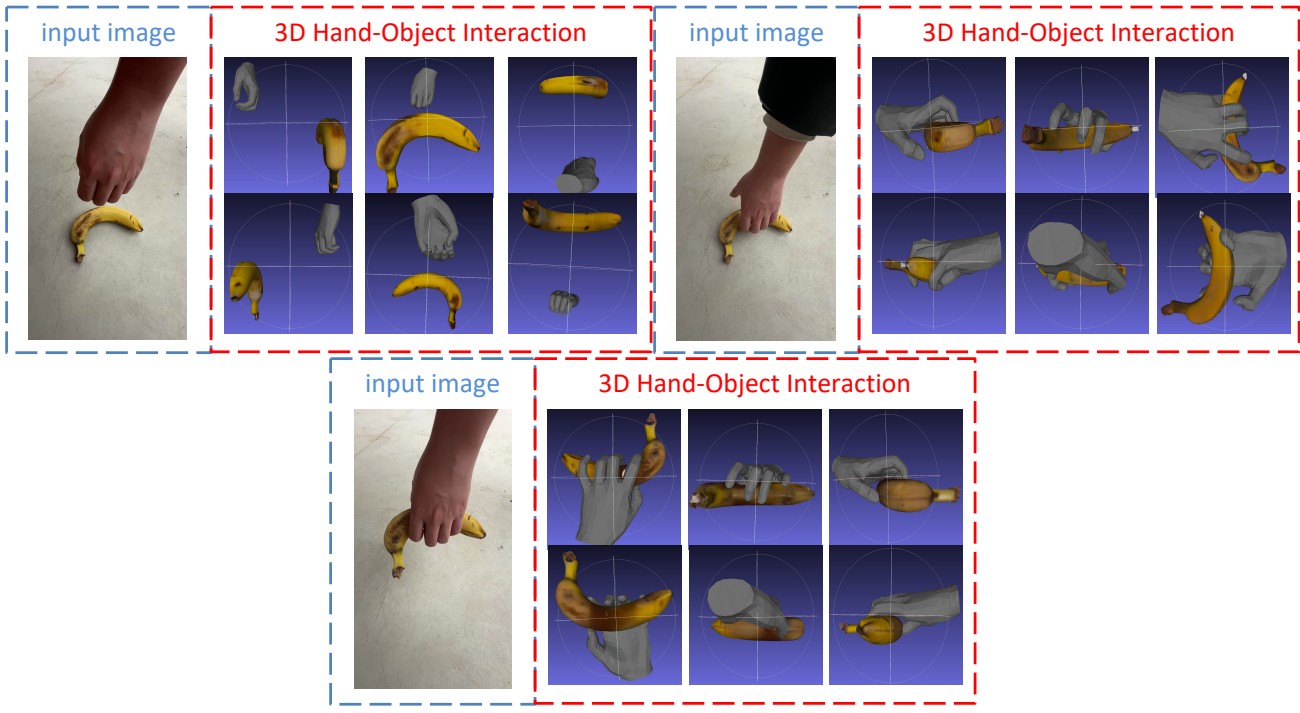

Figure 6: Visualizing the input image and the output hand-object interactions in 3D mesh.

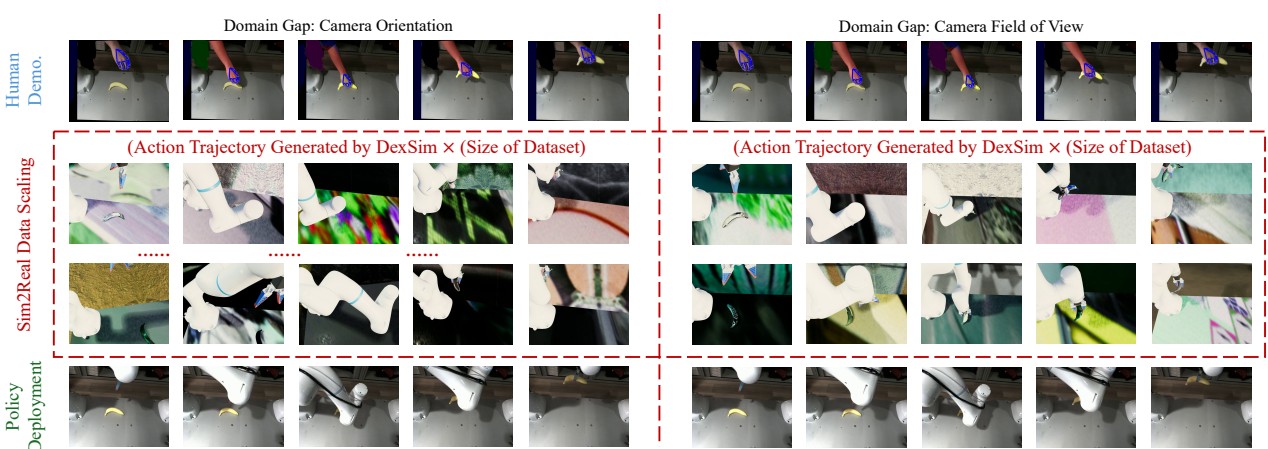

Figure 7: Visualizing the input demonstration (top row), examples of the scaled data (middle rows), and the realistic deployment of the robot policy (bottom row). The Sim2Real domain gaps are camera orientation (left) and field of views (right).

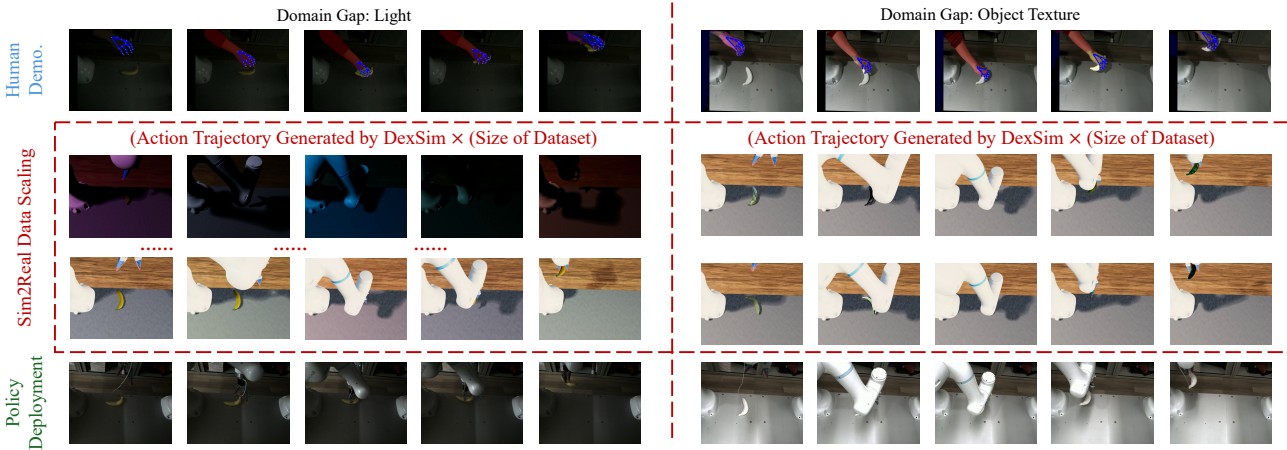

Figure 8: Visualizing the input demonstration (top row), examples of the scaled data (middle rows), and the realistic deployment of the robot policy (bottom row). The Sim2Real domain gaps are lighting (left) and object texture (right).

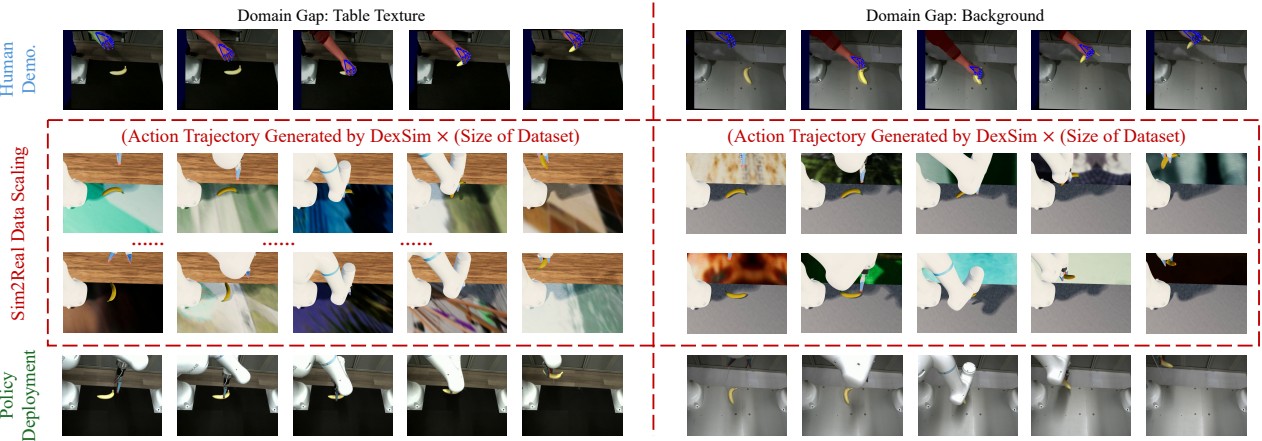

Figure 9: Visualizing the input demonstration (top row), examples of the scaled data (middle rows), and the realistic deployment of the robot policy (bottom row). The Sim2Real domain gaps are table texture (left) and background (right).

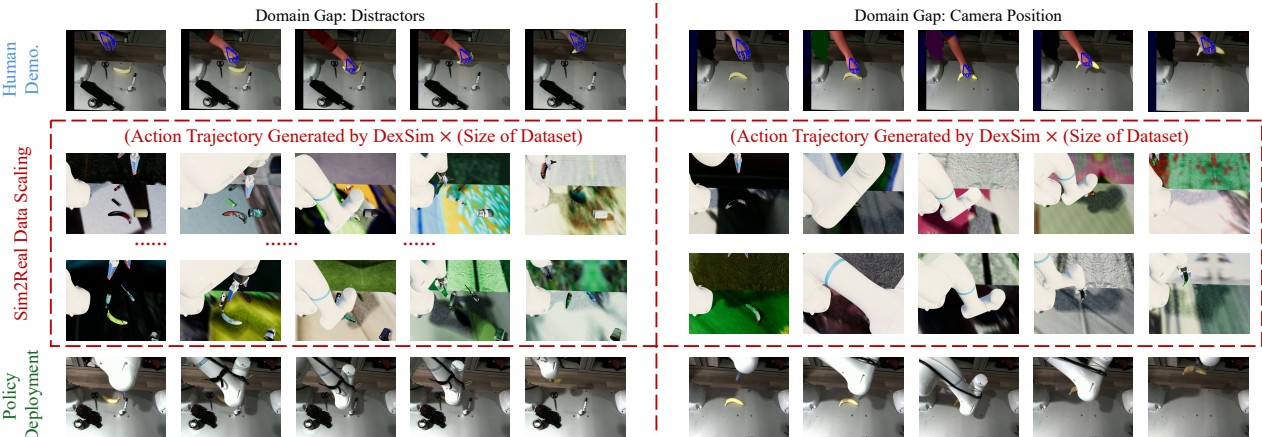

Figure 10: Visualizing the input demonstration (top row), examples of the scaled data (middle rows), and the realistic deployment of the robot policy (bottom row). The Sim2Real domain gaps are distractors (left) and camera position (right).

