# OpenReview forum: "DexScale: Automating Data Scaling for Sim2Real Generalizable Robot Control"
_ICML.cc/2025/Conference — ICML 2025 poster_

### Official Review · Reviewer_dQaj · 2025-03-07

**Overall Recommendation:** 3

**Summary:**

The paper introduces a data engine that automatically simulates and scales skills for learning robot manipulation policies. In particular, DexSim presents a comprehensive pipeline for Sim2Real data scaling by automating domain randomization and adaptation processes. The authors claim that this approach not only achieves superior zero-shot Sim2Real performance but also enhances generalization across diverse tasks.

**Claims And Evidence:**

**The paper claims:**
- DexSim can generate simulation trajectories based on real-world human demonstrations.
- DexSim facilitates improved zero-shot Sim2Real transfer for manipulation policies.
- DexSim supports generalization across a variety of tasks and multiple robot embodiments.

**However, several claims appear weak and are not well supported:**
- **Manual Adjustments and Pose Capture:**
  As illustrated in Figure 3, substantial manual adjustments are required to align the hand and object models within the simulation before retargeting them to the end-effector pose. Additionally, it remains unclear how the delta-pose is extracted from human hand video. Figure 7 further suggests that this approach is labor-intensive and may not be scalable.

- **Statistical Significance:**
  The performance differences between simulation and real-world results for skills+DR and DexSim are minimal. With additional experiments and the inclusion of standard deviation data, these differences might not be statistically significant.

- **Generalization with Perturbation:**
  The improvements in generalization achieved through perturbation appear marginal. A more robust evaluation—potentially using large-scale simulation environments such as the Factor-world or The Colossuem Benchmark, which incorporate perturbation variations—would strengthen this claim.

**Essential References Not Discussed:**

No

**Experimental Designs Or Analyses:**

The experimental design is sound. However, there are alternative methods for improving sim-to-real transfer—such as ASID, SystemID, and GenAug—that the authors could consider for comparison.

**Methods And Evaluation Criteria:**

The evaluation criteria are well-suited for assessing zero-shot sim-to-real transfer. However, the method itself is problematic due to its many moving parts, where a failure in any single component—given its reliance on other models—can undermine the entire process. Consequently, scaling this approach appears infeasible.

**Other Comments Or Suggestions:**

Nill

**Other Strengths And Weaknesses:**

Figure 1 does not effectively convey the core idea of the work, and the alignment of arrows and boxes appears to have been executed hastily.

Figure 2 includes numerous elements that are not adequately explained in the caption, which hinders the reader's understanding of the essential pipeline.

**Questions For Authors:**

Nill

**Relation To Broader Scientific Literature:**

Yes, the work could be more valuable with a broader discussion of existing sim-to-real methods beyond just domain randomization (DR) and domain adaptation (DA).

**Theoretical Claims:**

Yes, the problem formulation make sense.

---

> ### Author Rebuttal · Authors · 2025-04-01
>
> Dear Reviewer, we sincerely appreciate your constructive feedback. We hope that the following response can address your concerns:
>
> > 1. "*...substantial manual adjustments are required to align the hand and object models within the simulation before retargeting them to the end-effector pose. ... Figure 7 further suggests that this approach is labor-intensive and may not be scalable.*"
>
> DexSim features an automated pipeline for extracting human hand poses from videos and aligning them with objects **without any manual intervention**. The joint optimization process (see line 222, right column) between the hand and object models is fully automated. Specifically, similar to Liu et al. (2024), we utilize a learning-based approach to iteratively refine and synchronize hand-object interactions over multiple rounds until they are properly aligned. We have clarified this in the revised version of our study.
>
> (Liu et al., 2024)  Liu, Yumeng, et al. "EasyHOI: Unleashing the Power of Large Models for Reconstructing Hand-Object Interactions in the Wild." arXiv preprint arXiv:2411.14280 (2024).
>
> > 2. "*...The performance differences between simulation and real-world results for skills+DR and DexSim are minimal ...*"
>
> Thank you for pointing this out. In our experimental results, we observed that the performance gap between Skill+DR and DexSim is less pronounced in the object grasping task. One major contributing factor is the limited number of trials conducted in both real-world and simulated environments. To address this, we have increased the number of experiment runs. The updated results are shown below:
>
> | Setting | Skill | Skill+DR | Skill+DA | DexSim |
> | -------- | ------- | -------- | ------- | ------- |
> | Real-World | 3/25 | 7/25 | 9/25 | 14/25 |
> | Simulation | 93/200| 122/200 | 151/200 | 157/200 |
>
> To further support our findings, we have included the relevant experiment videos in Section 3.1 of our [anonymous link](https://anonymous.4open.science/w/dexscale/).
>
> > 3. "*A more robust evaluation—potentially using large-scale simulation environments such as the Factor-world or The Colossuem Benchmark, which incorporate perturbation variations—would strengthen this claim.*"
>
> Our study focuses on real-world experiments, demonstrating improvements in generalization performance on physical robots rather than in simulated environments. Specifically, our experiment on generalizability (Section 5.1) addresses the challenge of bridging the Sim2Real gap. We show that the automatic domain randomization (DR) and domain adaptation (DA) mechanisms in DexSim effectively transfer policies learned in simulation to real-world applications.
>
> In contrast, benchmarks that study generalization across discrepancies between training and testing environments, such as the Factor-World or the Colosseum Benchmark, **still evaluate performance in simulated settings**. However, as we argue in Section 3, the Sim2Real gap presents a significant challenge. There is no guarantee that a policy performing well in these benchmarks can be successfully deployed in real-world scenarios.
>
> > 4. "*The method itself is problematic due to its many moving parts, where a failure in any single component can undermine the entire process.*"
>
> The term "moving parts" in the review seems confusing. Generally, in methods implementing the Real-to-Sim-to-Real pipeline, the presence of multiple components is a fundamental characteristic. In the case of DexSim, these components are not only necessary but also integral to the pipeline's functionality. Similarly, methods like RoboGen and GenSim also adopt pipelines composed of multiple interconnected parts. Therefore, the presence of "many parts" is not unique to this method but rather a common feature of such approaches.
>
> > 5. "*... there are alternative methods for improving sim-to-real transfer—such as ASID, SystemID, and GenAug...*"
>
> While these studies are indeed inspiring, we found that their methods are not directly comparable to ours. Specifically:
>
> - ASID focuses on refining simulation models to better represent real-world dynamics. In contrast, DexSim interacts with a given dynamics model without access to its underlying function.
> - SystemID addresses system identification by recovering nonlinear models of dynamical systems from data. However, it does not consider simulated environments or skill discovery methods relevant to our study.
> - GenAug appears to be a data augmentation technique leveraging image-text generative models. Notably, its augmentation approach enforces action invariance. In contrast, DexSim can autonomously refine actions when the target object changes.
>
> > 6. "*Figure 2 includes numerous elements that are not adequately explained in the caption.*"
>
> The elements in the figures have a one-to-one correspondence with the titles of the subsections and paragraphs in Section 4: Data Engine for Sim-to-Real Generalization. This relationship has been clarified in the revised version of our study.

---

### Official Review · Reviewer_ioHD · 2025-03-13

**Overall Recommendation:** 2

**Summary:**

The paper proposes a new data generation pipeline, that takes human video demo as input and generate retargeted data for robot manipulation. The pipeline involves different stages: scene projection, action-trajectory projection, scene simulation, action-trajectory simulation and various techniques to bridge the sim2real gap. The author conducts experiment on simple pick-and-place and open-box problems to suggest the proposed data generation pipeline produces better data quality to train manipulation policies.

**Claims And Evidence:**

The paper claims that the pipeline is more powerful in generating data. However, I only see simple tasks like grasping and open a box. These tasks are relatively fault tolerant comparing more dextrous task like opening a cabinet door via the handle. It is unclear whether the action retargeting algorithm will still provide useful trajectories to solve the problem.

**Essential References Not Discussed:**

Discussion with paper like DexMimicGen: Automated Data Generation for Bimanual Dexterous Manipulation via Imitation Learning, where it is also aims for synthetic data generation but with different types of inputs.

**Experimental Designs Or Analyses:**

The experiment is supportive for the dexsim techniques on the given tasks. However, as mentioned above, it fails to support that dexsim is a valuable platform to generate valuable synthetic trajectories for other dexterous tasks, which is of ture interest of the community. For simple pick and place tasks, one can also design procedural ways to generate synthetic data easily.

**Methods And Evaluation Criteria:**

The physical part is build on a custom physics engine in the current implementation. Why not using the current available platform like mujoco / pybullet?

For the scene / action projection, i would like to understand the impact of out-of-domain cases where one has to find similar objects in the set. How will this effects the quality of the synthetic data?

**Other Comments Or Suggestions:**

None

**Other Strengths And Weaknesses:**

None

**Questions For Authors:**

1. Why not using the current available platform like mujoco / pybullet?
2. How would out-of-dataset objects effect the retarget trajectories?
3. Could you provide evaluation on more dexterous tasks?

**Relation To Broader Scientific Literature:**

The paper proposes a way to generate synthetic data given input of real-world human demos. This aims to tackle the challenge in robot learning, where such data is scarce in practice.

**Theoretical Claims:**

There is no theoretical claims.

---

> ### Author Rebuttal · Authors · 2025-04-01
>
> Dear Reviewer, We sincerely appreciate your constructive feedback and thank you for recognizing the significance of our work. We have carefully considered your suggestions, and we hope that the following response can address your concerns:
>
> > 1. *"...I only see simple tasks like grasping and open a box. These tasks are relatively fault tolerant comparing more dextrous task like opening a cabinet door via the handle. Could you provide evaluation on more dexterous tasks?"*
>
> Our experiment comprises three distinct tasks, each involving a unique embodiment. To further support our argument—and in response to the reviewer's request—we additionally demonstrate DexSim's performance on two representative tasks: articulated object manipulation (e.g., opening a drawer) and combined grasping and manipulation (e.g., water pouring). Detailed results and examples can be found in Section 2.1 of our [anonymous link](https://anonymous.4open.science/w/dexscale/).
>
> Additionally, we emphasize that the tasks demonstrated in our experiment are inherently challenging. For instance, the box-opening task requires the robot to sequentially open four flaps, demanding precise control and planning to complete the task successfully.
> Even more complex is the re-arrangement task, where the robot must reorient both the fork and spoon so they face the front of the plate, while also positioning them accurately around it. Smooth execution of this task necessitates coordinated movement between both arms, ensuring that the utensils are placed correctly and simultaneously.
>
> > 2. "*Why not use existing platforms like MuJoCo or PyBullet?*"
>
> Choosing a simulation platform involves balancing several factors: rendering quality, physics accuracy, overall simulation efficiency, and the flexibility and scalability of the simulator.
>
> Unfortunately, we have not found an existing open-source simulator that satisfies all of these requirements. For example, PyBullet and MuJoCo offer strong physics simulation capabilities, but their rendering quality is limited. On the other hand, Blender excels at rendering but is not designed for physics simulation—especially not for robotics applications.
>
> Recently, several simulators have emerged for robotics and embodied AI, such as AI2-THOR, RobotSuite, and SAPIEN/ManiSkill, but each comes with notable limitations:
>
> AI2-THOR uses Unity3D as its simulation backend. While Unity is a well-known game engine, it lacks support for GPU-parallel simulation and tiled rendering, which are crucial for large-scale robotics experiments.
> RobotSuite separates the physics and rendering engines and integrates them via a Python frontend. This architecture leads to inefficiencies and does not scale well for large simulations.
> SAPIEN/ManiSkill supports GPU-based simulation and tiled rendering, but it only handles rigid-body dynamics and lacks the flexibility needed to extend or incorporate new features.
>
> IsaacSim/IsaacLab comes closest to meeting all our requirements. However, it is not fully open-source, which limits our ability to modify or extend its low-level capabilities to fit our specific needs.
>
> > 3. "*How would out-of-dataset objects effect the retarget trajectories?*"
>
> DexSim includes an automatic domain randomization method to address mismatches between objects in simulated and real-world scenes. Specifically, the reviewer pointed out in the Scene Projection section (Section 4) that the targeted object is included in the asset dataset (OoD), and DexSim matches it with a similar object. Consequently, the skill is learned based on this matched object rather than the actual target object.
>
> However, because our domain randomization process modifies the size, shape, pose, and texture of objects, the learned skills can generalize across different objects. As a result, the model can successfully handle the real object, even if it differs from the one used in the simulation. We have experimentally validated this phenomenon in Section 5.1.
>
> > 4 "*Add Discussion with paper like DexMimicGen: ...*"
>
> We have included this paper in our related works and discussed it in detail. The key differences lie in the format of input signals, the methods for skill generation, and the integration of automatic domain randomization and adaptation for Sim2Real generalization.

---

### Official Review · Reviewer_Fjdp · 2025-03-16

**Overall Recommendation:** 3

**Summary:**

This paper proposes DexSim, a pipeline for automating the learning of manipulation skills from human videos. Given an egocentric video, it first extracts human hand, wrist, and object pose trajectories. It then retargets the hand trajectory to a robot gripper and finds an object mesh closest to the object in the video. After that, it builds a simulation environment with objects, layouts, and scenes, and uses an LLM to randomize scene configurations. In this randomized environment, it then generates trajectories as the ground-truth demonstrations and learns a policy for deployment in the real world.

**Claims And Evidence:**

The paper makes several claims regarding the effectiveness of the proposed method, but not all are well supported:

1. Scalability across different embodiments: The paper claims scalability, and while a few supplementary videos show different robots, there is no quantitative comparison in the main text or supplementary material. More analysis is needed in this part.

2. Effectiveness of automated domain randomization (DR) and domain adaptation (DA): The experiments show that these techniques improve Sim2Real transfer, but additional ablations are necessary. For example, how does automated DR compare with hand-crafted DR? What is the contribution of individual components (e.g., AI-DR vs. SA-DR)? Similarly, what are the performance gains from object-oriented representations and pose-affordance representations in DA?

3. Unverified claims in figures: Some figures include elements that are not backed up by experiments. For instance, Figure 2 mentions image/teleoperation as data sources and RL for policy fine-tuning, but these are not used in the experiments. While it is reasonable to illustrate pipeline flexibility, it may mislead readers if these elements are not actually part of the method.

**Essential References Not Discussed:**

The paper covers most relevant references.

**Experimental Designs Or Analyses:**

I checked the validity of all experiments, from the experimental design to the results. The existing designs are valid but would benefit from further analysis, especially regarding:

1. Quantitative analysis across different embodiments.
2. More detailed ablation experiments.

(See Claims and Evidence for further details.)

**Methods And Evaluation Criteria:**

The proposed method is reasonable for the problem setting.

**Other Comments Or Suggestions:**

No additional comments. Please see my detailed points above.

**Other Strengths And Weaknesses:**

Most of my concerns are expressed below, here are some additional points:

(+) The paper addresses an important challenge in Sim2Real learning and proposes an automated and scalable pipeline.

(-) It lacks detailed ablation experiments on the effectiveness of different components.

(-) This paper does not show quantitative results for different embodiments.

(-) I find the current paper presentation can be improved. Some figure elements (e.g., teleoperation, RL fine-tuning) are not reflected in the method section. It would be helpful if the authors explicitly stated what is implemented and what is a possible future extension. Additionally, clearer explanations on how human hand trajectories are mapped to robot hands and how mesh refinement is obtained from video capture would improve clarity.

**Questions For Authors:**

1. Similar to mentioned above, what are the performance contributions of each module? And what are the performance for different
2. How do you implement the retargeting from human hand to different robot embodiment?
3. What are the primary failure modes observed in real-world deployments?

**Relation To Broader Scientific Literature:**

The paper builds upon prior work in Sim2Real transfer, domain randomization, and imitation learning. The discussion of related work is thorough, but a detailed comparison to baselines (e.g., naive randomization) would clarify the contribution.

**Theoretical Claims:**

No theoretical claims are presented in this paper.

---

> ### Author Rebuttal · Authors · 2025-04-01
>
> Dear Reviewer, we sincerely appreciate your constructive feedback. We hope that the following response can address your concerns:
>
> > 1 "*Only a few supplementary videos show different robots, there is no quantitative comparison in the main text or supplementary material.*"
>
> **Response.**
> Table 4 in the appendix summarizes the various embodiments used to demonstrate their performance in real-world scenarios.
> To better support our claim, we experiment with an additional robot for grasping and manipulating (e.g., grasping a bottle and pouring the liquid into a cup). This experiment is conducted on a dual-armed Cobot Magic robot, i.e., mobile ALOHA (Fu et al., 2024).
> The success rates are as follows:
>
> | Setting | Skill | Skill+DR | Skill+DA | DexSim |
> | -------- | ------- | -------- | ------- | ------- |
> | Real-World | 1/10 | 2/10 | 3/10 | 7/10 |
> | Simulation | 30/100| 54/100 | 71/100 | 86/100 |
>
> For the relevant videos for different embodiments, please check section 1.1 in our [anonymous link](https://anonymous.4open.science/w/dexscale/)
>
> Moreover, previous studies on the Sim2Real robotic simulation data engine primarily focus on a single type of embodiment. In contrast, our real-world experiment is conducted on a significantly larger scale.
>
>
> > 2. "*How does automated DR compare with hand-crafted DR?*"
>
> Handcrafted DR refers to selecting the type and scale of DR based on human expertise. We refer to the empirical analysis by Xie et al. (2024), which ranks the importance of various DR features. To evaluate model performance, we apply the top 1, 2, and 3 most important DR features to bridge the Sim2Real gap and report the results as follows.
>
> | Setting | Camera Orientation | Camera Orientation + Table Texture | Camera Orientation + Table Texture + Distractors | DexSim |
> | -------- | ------- | -------- | ------- | ------- |
> | Real-World | 1/10 | 3/10 | 4/10 | 7/10 |
> | Simulation | 62/100| 63/100 | 73/100 | 86/100 |
>
> For the detailed videos, please check section 1.2 in our [anonymous link](https://anonymous.4open.science/w/dexscale/) for detailed results.
>
> (Xie et al., 2024) Xie, Annie, et al. "Decomposing the generalization gap in imitation learning for visual robotic manipulation." ICRA 2024.
>
> > 3. "*What is the contribution of individual components (e.g., AI-DR vs. SA-DR)? Similarly, what are the performance gains from object-oriented representations and pose-affordance representations in DA?*"
>
> Regarding the comparison between AI-DR and SA-DR, their configurations are automatically determined by the LLM, and as such, there is no guarantee that they will always be included in the DR variants generated by DexSim. This makes it challenging to isolate and evaluate their individual effectiveness in empirical studies. Therefore, we choose to study DR as an integrated component in our analysis.
>
> The same argument applies to object-oriented representations and pose-affordance representations in DA. There is no guarantee they will always be included in DexSim.
>
>
> > 4. "*Figure 2 mentions image/teleoperation as data sources and RL for policy fine-tuning, but these are not used in the experiments.*"
>
> We apologize for the misunderstanding. Our Cobot Magic robot is equipped for teleoperation. Since the teleoperation signals are directly compatible with the robot, retargeting is unnecessary. Instead, we can directly transfer these control signals to the simulator to control the simulated robot within the data engine.
>
> Regarding RL fine-tuning, DexSim supports the automatic design of reward and goal functions by leveraging large language models (see lines 270–273, right column). Given these reward functions, applying RL algorithms to DexSim is straightforward.
> Well-established algorithms, such as PPO and SAC, can be seamlessly integrated into our engine to generate skills.
>
> > 5. "*The reorientation task appears to be a simple pick-and-place...*"
>
> The re-arrangement task is significantly more challenging than simple pick-and-place operations for the following reasons:
> 1. The robot arm must reorient the fork and spoon to ensure they face the front of the plate and position them correctly around it.
> 2. To achieve smooth execution, both arms must work in coordination, ensuring the fork and spoon are placed in their correct positions simultaneously.
>
> > 6. "*What are the performance contributions of each module?*"
>
> In section 5.1, we have conducted an ablation study by removing either the strategic Domain Adaptation (DA) or Domain Randomization (DR) components from our DexSim dataset. We have also added an experiment for comparing different DR methods (either hand-crafted or automated) as mentioned above.
>
> > 7. "*What are the primary failure modes observed in real-world deployments?*"
>
> We have included a fail case study in the revised draft. Please see examples in section 1.3 in our [anonymous link](https://anonymous.4open.science/w/dexscale/) for detailed results.

---

### Decision · Program_Chairs · 2025-05-01

**Decision:**

Accept (poster)

**Comment:**

The paper introduces DexSim, a tool for generating diverse large-scale demonstration data for robotic manipulation tasks in simulation based on human videos.

DexSim targets one of biggest problems in robot learning, data scarcity. The reviewers appreciate its significance, as well as DexSim's potential for alleviating it.

Originally, the reviewers had a number of concerns regarding DexSim's empirical evaluation, including the effectiveness of its domain randomization. However, the experiments done by the authors for the rebuttal have largely addressed these. The two remaining issues with this work are:

**Writing/presentation quality.** The reviews mention the specific details. None of these issues are grave, and all can be fixed in the camera-ready. The metareviewer strongly encourages the authors to do so.

**The limitations in tasks for which DexSim can generate data.** DexSim is largely limited to pick-and-place tasks. While the authors claims it goes beyond that and can handle tasks such as pouring and reorientation, the videos provided by the authors in the rebuttal don't support these claims (e.g., no liquid is actually being poured in the pouring task videos). Tasks that involve assembly, such as screwing a nut onto a bolt, also appear to be outside DexSim's scope. Nonetheless, this limitation is understandable and stems, among other factors, from the limitations of the underlying simulators. Other robot data generation frameworks such as DexMimicGen, are even more restrictive.

Thus, on balance the metareviewer believes that the paper makes a very valuable contribution to solving the data challenges of robot learning.